# Scatterpixels: Ad Hoc Reconfigurable Physical Pixel Displays

Antony Albert Raj Irudayaraj*    Jeremy Hartmann†    Omid Abari‡    Daniel Vogel §

Cheriton School of Computer Science, University of Waterloo

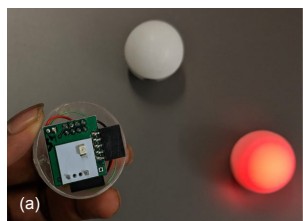 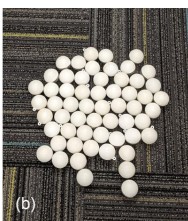 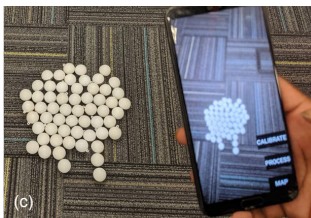 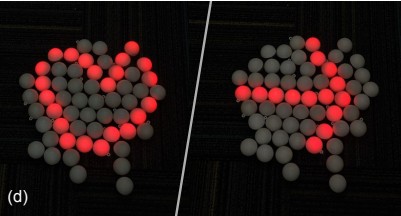

Figure 1: Scatterpixel system: (a) each 4cm spherical "pixel" is an independently addressable unit with a rechargeable battery, microprocessor, and radio to control a red LED; (b) one example usage where pixels create an ad hoc floor display; (c) the relative spatial locations of the pixels are registered through computer vision using a smartphone application; (d) once registered, rendering algorithms display patterns or symbols at optimal positions using a communication protocol capable of 20 FPS animation.

## ABSTRACT

Computer displays are a matrix of small, precisely aligned pixels to maximize fidelity within a standardized, defined area. We explore an alternative concept where pixels are individual physical entities that can be physically positioned and combined to create temporary ad hoc displays with arbitrary shapes, sizes, and functions. The fabrication and hardware design for a spherical single LED pixel is described that is inexpensive, wireless, and rechargeable with more than 70 pixels assembled. Smartphone-based computer vision methods are developed to register the positions of addressable pixels in arbitrary arrangements, and interactively guide fine-tuning of pixel positions when display content is known. Once registered, algorithms are presented to optimally render content on the pixels, and a communication protocol enables frame rates suitable for animation. Example pixel display configurations are described for applications like audience participation, way-finding, dynamic event signage, time-keeping, scoreboards, and ambient displays.

**Index Terms:**

Human-centered computing—Human Computer Interaction (HCI)—Interaction devices—Displays and Images

## 1 INTRODUCTION

Traditional computer displays are a matrix of small, precisely aligned pixels that maximize fidelity within a standardized, defined area. We explore an alternative visual display concept called *ad hoc reconfigurable displays* where pixels are individual physical entities that can be positioned and combined to create re-usable displays with arbitrary shapes, sizes, and functions. The goal is to create a portable and easy-to-use system that trades the fidelity of conventional displays for a high level of flexibility in display configuration, enabling new kinds of ubiquitous display use cases and novel aesthetic display experiences. For example, the same physical pixels can be reconfigured to form a long corridor display to nudge people to a meeting room in the morning, attached to a wall to show the score

*e-mail: aariruda@uwaterloo.ca

†e-mail:jhartmann@uwaterloo.ca

‡e-mail: omid.abari@uwaterloo.ca

§e-mail:dvogel@uwaterloo.ca

of a game of basketball in the afternoon, and scattered on a lawn to create a fun welcome sign for a party in the evening.

Previous work has explored different approaches to enabling related types of reconfigurable displays. For example, using many mobile phones or tablets held by people in a crowd where each device acts as a pixel in a large temporary display [37, 41, 47]. A compelling concept, but tailored to a specific setting and use case. Miniature robots or drones have been created to arrange themselves as pixels in dynamic displays (e.g. [8, 19, 43]), but they are complex, expensive, have high power requirements, and require an instrumented environment for continuous tracking. They are well suited to applications that benefit from the capabilities of a realtime dynamically reconfigurable display.

We focus on a different problem space for temporary *manually reconfigurable displays*, and target use cases and applications that are impractical or undesirable to create with a swarm of robots. For example, a one-dimensional way-finding display created along a long corridor during an afternoon of meetings, or a sign for a party placed on a grassy lawn for an evening. The most related approaches to our work are systems that use addressable LEDs as pixels [4, 6, 12, 13, 36, 37, 42]. But these systems remain incomplete: some use approaches that rely on hard wiring for either communication or power; some use communication protocols not capable of real-time animations; and many are limited in the method and versatility for how pixels are spatially registered. *No previous work has holistically examined all technical aspects necessary to realize the potential of using individual LED pixels for reconfigurable displays.*

This paper describes Scatterpixels, a comprehensive, and flexible, and usable reconfigurable ubiquitous display system. The core of the system is a 4 cm spherical "physical pixel" using a single red LED (Figure 1a) that is inexpensive, wireless, rechargeable, and can be produced at scale. Many of these pixels can temporarily be distributed along a corridor, attached to a ferromagnetic surface, fastened to a wall or window, or spread out on the floor or ground (Figure 1b). Smartphone-based computer vision methods register the positions of addressable pixels in arbitrary arrangements (Figure 1c), including providing interactive augmented reality guidance so the operator can fine-tune pixel positions. Once registered, algorithms optimally render content on the available pixel layout (Figure 1d,e). A compact and fast communication protocol enables animation frame rates on the 70 pixels we fabricated, and will scale to hundreds of pixels. Using our system, we demonstrate different display configurations and applications, ranging from individual visual indicators, to one-dimensional way-finding, to two-dimensional

dynamic event signage.

In summary, we contribute a complete end-to-end reconfigurable display system using faster, smaller, more self-contained hardware, more comprehensive registration methods enabling a greater variety of display configurations, new methods to fine tune pixel layouts for specific content, and integrated techniques that optimize content for a given layout. Our methods are packaged in a way to make the complete system usable: simple deployment requiring only a base station, phone, and remote server; a convenient charging method; and a phone-based app to control all functions. Our open source schematics, design files, and code are all available[1].

## 2 RELATED WORK

Before providing details of our system, we discuss how it relates to previous ad hoc reconfigurable display approaches. We recognize that definitions of a reconfigurable display could include systems that actuate or combine conventional display units. For example, using actuated projectors to form displays on different surfaces [10,32,46], creating a modular set of back-projection display "bricks" [26,39], combining multiple device screens together to create large high resolution displays, [23, 28, 34, 40], robotic large display panels that "shape shift" [44], even using drones with projectors to create flying high resolution displays [18,31,38]. However, our focus is on previous work that explicitly, or conceptually, considers each display pixel as an independent and distinct element that can be re-positioned to create different displays.

### 2.1 Using many phones or tablets as pixels

One possible approach in some settings, is to create a large ad hoc display using many phone or tablet screens as individual pixels. Schwarz et al. [21, 41] and Chungkuk Yoo et al. [47] investigate variations of this general idea, where many people in a crowd run a special application which communicates a unique code (such as flashing different colour transitions) to a centrally positioned camera, so the relative location of all phones can be determined. Once registered, and assuming everyone maintains a similar pose, imagery and patterns can be presented on the display created by a collection of phones, each acting as a single pixel. This type of display is truly ad hoc, meaning there is little a priori control of its shape or how it might change over time. However, the usage setting and possible configurations are not general, and using one phone per pixel may not be practical due to size and cost.

### 2.2 Using drones or robots as pixels

If dynamic deployment and realtime control of the display shape is important, independent physical actuation of pixel elements is possible. Bitdrones [8] is an actuated 3D display using nano-quadcopter drones, demonstrations used up to 12 drones each acting as a single RGB pixel. The system highlights methods for real time tracking and absolute position control when faced with challenging conditions resulting from many small drones flying close together. Bitdrones can only create sparse displays since drones can not fly too close to each other due to turbulence and downdraft. The system requires a high quality Vicon motion tracking setup in the environment, and each drone can only run for about 7 minutes requiring frequent recharging. Commercial groups have used many drones to create dynamic outdoor displays [1]. These are intended for large public events since significant planning and setup with a team of people is required, and there are still limitations for run time and display density. However, the outdoor setting enables the use of a new generation of error-corrected GPS positioning.

Tangentially related to our work are systems that use small robots, each acting as a single pixel that can be actuated to form dynamic two-dimensional displays. Morphogenesis [43] uses circular robots

called kilobots [35] that contain a battery, vibrating motors, multi-color LED, and an IR LED to communicate with neighbouring pixels. Communication is line-of-sight with an operating range of 10cm. The system does not centralize the control of robot positions to form specific shapes for a conventional information display. Instead, the robots self-arrange in periodic patterns or shapes to mimic phenomena like cell behaviour in tissue growth.

Other systems centrally control and track individual wheeled robots. For example, PixelBots [3] use an overhead camera, Hiraki et al.'s robots [11] decode invisible projected light patterns [15] from a specialized projector, and Zooids [19] use a related method of detecting projected grey-code patterns. These robots can be compact, for example Zooids [19] are 2.6cm diameter cylinders. Our system uses the same radio transceiver module, voltage regulator, and battery charging chip as in Zooids.

A reconfigurable display composed of self-actuated robot pixels is well suited for applications like physical animations or dynamic optimization of layouts. Indeed, a central focus of these works are algorithms to optimally re-arrange robots to convey a given image or display dynamic animations [2]. However, self-actuation has a significant trade-off with cost, complexity, and flexibility. An instrumented environment is required for accurate registration and continuous tracking (dead reckoning remains a difficult problem and self-localization is limited by physical constraints like line-of-site). In some cases, the increased time to perform actuated movements may detract from the display. In many cases, miniature robots will not work diverse surfaces without specific customization, for example, on a carpet or grass. A manually reconfigurable display trades self-actuation capability for a system that is simpler, less expensive, and useful for a different set of applications across different display form factors. In our system, pixels are positioned by hand, possibly with the aid of a smartphone app, and the same app is used to register their static location.

### 2.3 Using individual stationary LEDs as pixels

There are several examples of art displays composed of physically separate pixels. LED Throwies [45] are individual powered LEDs that can be attached to ferromagnetic surfaces to create sparse, abstract displays by manually positioning them. They are always on until the battery drains. Six-forty by four-eighty [5] is an art installation with 220 individual pixel "blocks." Each has a small screen to display animations and colours when touched. Display-Blocks [33] are cube-shaped pixels with small OLED screens one each side. Each cube displays images or videos independently, from data pre-loaded onto an internal SD card. None of these examples use pixels that are spatially registered and none feature communication between pixels, so creating coordinated, dynamic displays is not possible.

#### 2.3.1 Physically connected with fixed layouts

Other examples connect individual pixels by wires to enable communication, but use fixed or manual spatial configurations. Lightset [12] hangs chains of wired LEDs on exterior building walls to prototype and explore ideas for urban displays. LED positions are known on each chain, and the demonstrated layouts are regular 2D grids created by multiple chains, both of remove the need for custom registration. The distance between pixels can only be adjusted between 5 to 30 cm because of wiring, making this approach unsuitable to create diverse display layouts. Sato et al. [37] create large displays for an airport ceiling by arranging individual LEDs to display imagery like stars and simple animations. This semi-permanent, purpose-built installation places each LED at a pre-computed position according to the specific ceiling site.

---

[1]https://github.com/exii-uw/scatterpixels

### 2.3.2 Physically connected and reconfigurable

More related are systems which form larger images using individual display modules as a "pixel". Siftables [29] are 36mm square "tiles" each containing a 128 by 128 px colour LCD display, IR transceiver to communicate with nearby tiles within a 1 cm range, and RF modules to communicate with a central base station. These can be arranged to form larger displays in rows or grids with each tile rendering a section of the combined image or application. Pickcells [7] is a similar system composed of small modular colour LCD screen tiles created from commercial smartwatches that can be physically connected to form different shapes. Both projects demonstrate using each tile to render an image, or a piece of a larger image. In theory, a large number of tiles could be connected to form a very large display with each LCD tile forming single RGB pixels, but this was not the focus and was not tested or demonstrated.

### 2.3.3 Wired and reconfigurable

Yet other examples use physically connected or wired "pixel" modules and support some limited forms of registration. Pushpin [24,25] is a modular system for designing table-top wireless sensor networks composed of nodes built by stacking individual 18×18mm modules for power, communication, processing, and application-specific functions like an LED or light sensor. Nodes are centrally programmed by an IR spotlight with communication among nodes using capacitive coupling or IR LEDs. Each node is powered from a common "power plane" of layered aluminum foil and polyurethane foam, which reduces reconfigurability to the area of the plane. Blinky [16] are 40mm cube-shaped physical blocks each containing multiple RGB LEDs to produce the same colour in all directions, an orientation sensor, and contact connectors to communicate with neighbouring blocks. Multiple blocks can be reconfigured into rectilinear shapes by connecting them in lines and stacks. The kind of displays demonstrated are limited since the focus is on using the system to teach programming concepts.

Twinkly [13] is a commercial product for creating light decorations. It uses a string of LEDs or lamps wired together in configurations suitable for different scenarios, like seasonal decorations for a building exterior or enhanced Christmas tree lighting. Multiple strings are interfaced with a controller, which communicates with a phone. Calibration uses the phone camera, but the method is not specified. Firefly [4] is a semi-permanent display formed using hundreds of individually addressable lighting elements. Each lighting element contains a microcontroller and an LED, which connects to a common rail to obtain power and control signals. A sample display installation was formed by attaching 2940 lighting elements to a building exterior spread over $40m^2$ area. The spacing between the lighting elements can be adjusted to form arbitrary display configurations and accommodate for existing building infrastructure. The physical location of the pixels is obtained by decoding the flashing sequence of pixels using a centrally positioned high-quality camera. We use a similar method to register the spatial location of our pixels but support a combination of multiple captures for flexibility in deployment, camera requirements, and display shape. Wiring constraints between the pixels reduce the flexibility to create different display layouts and usage in different applications.

### 2.3.4 Wireless and reconfigurable

A more flexible approach is to design each LED pixel with on-board power and wireless communication. Bloxels [20] are translucent cube-shaped pixels which are stacked to form arbitrary-shaped volumetric displays. Each cube contains two RGB LEDs, 9 IR LEDs for communication, and a battery. Invisible light patterns are projected from under the table to communicate with the bottom Bloxels, with information passed to upper Bloxels using the IR LEDs. SteganoScan Orbs [17] are transparent spherical balls that can be rolled inside a large parabolic dish. Each ball contains 6 LEDs and 18 photo sensors, and they are tracked in real-time and the LED state updated by decoding invisible projected light patterns. The parabolic dish causes the orbs to pack close together when at rest to form a regular display grid. Urban Pixels [42] is an art installation to "paint building surfaces" with LED pixels. The battery-powered LED pixels are 4-inch acrylic balls, and they support wireless communication from a base station to display coordinated images and animations. The spatial location of the pixels are hard-coded, no spatial registration method is described. NetworkedPixels [6] create abstract light patterns across a large open garden area. The system is a network of 923 wireless LED nodes controlled asynchronously by a base station. Given the outdoor application and large distance between each nodes, onboard GPS is used for a simple spatial registration based on distance from the base station. Using the system as a single display to show coordinated imagery is discussed but not implemented. ParticleDisplay [36] uses 100 individually powered LED nodes(S-Node RFID module), each over 3cm square and 1cm thick, that can be controlled wirelessly from a base station. The base station communicates with the LED nodes at 4800 bps using a 303MHz radio module, which is unsuitable for displaying content in real-time or animations. The spatial location of the pixels is obtained using a simple method: a camera captures the entire display to record a video as each LED illuminates one-by-one in a known sequence. The LED node also contains an acceleration sensor to sense input and directly control it.

Our spherical pixel form factor and battery-powered wireless approach is most similar to Urban Pixels, but we reduce the size by more than a factor of 2, and we developed spatial registration methods that significantly advances the simple and constrained methods used by NetworkedPixels and Particle Display.

## 3 HARDWARE AND SYSTEM DESIGN

This section provides the hardware details for a "physical pixel," as well as associated parts of the Scatterpixels system for pixel charging and communication.

### 3.1 Physical Pixel

Each pixel (Figure 2b) is a 4cm diameter plastic sphere, which can opened into two hemispheres. The bottom half contains all components: a 25 by 20 mm 150mAh Lithium-ion rechargeable battery; a 23 by 24 mm custom PCB with an microcontroller (Atmega328), 3.3v voltage regulator (ADP122ACPZ), and other small components; a 2.4 GHz wireless transceiver board with a PCB trace antenna (NRF24L01+); and a 15 by 15 mm custom PCB board to mount a LED. The red LED is a $2.8 \times 3$ mm SMD (VLMS334AABB-GS08), with 1600 mcd luminous intensity and $\pm 60$ angle of half intensity). Two metal bulletin board thumb tacks, with flattened heads, are mounted through the bottom and side of the lower hemisphere to create battery charging contact pins.

The LED board is white to maximize reflection and designed to position the LED in the lower centre of the top hemisphere when the pixel is assembled. The top hemisphere is left empty and the inside is coated with a thin film of white spray paint to evenly diffuse the LED light. The outside of both hemispheres are coated with super matte transparent spray paint to eliminate specular reflections that would otherwise cause issues when computer vision methods are used for registration. The spherical shape of the pixel allows it to roll on surfaces and encourages more creative, less precise display layouts. A standard 1/4 inch hardware nut is placed inside the lower hemisphere at the very bottom to act as a ballast to further lower the centre of mass of the pixel. Each pixel weights approximately 17 g. This weight distribution, and slightly flat bottom profile created by the bottom charging pin, means when the pixel is placed or rolled on the floor, it eventually rights itself so that the LED diffuser half is up. The hardware nut can be replaced with a neodymium ring magnet

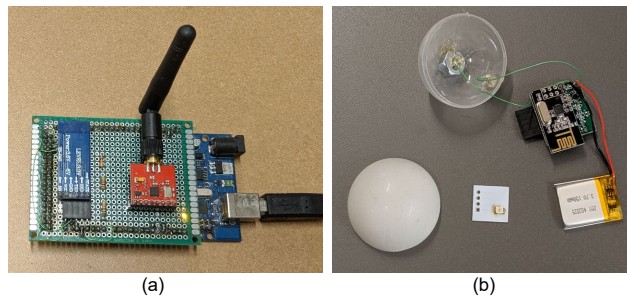

(a)          (b)

Figure 2: Hardware setup: (a) Bluetooth base station; (b) pixel disassembled into two halves showing primary components.

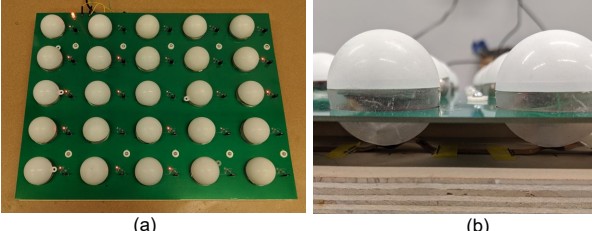

(a)          (b)

Figure 3: Pixel charging station: (a) 25 pixel charging board; (b) detail showing pixel contact with ground plane through bottom push pin and with charging circuit through conductive rim and side push pin.

of similar size, which provides enough attractive force through the bottom charging pin to attach the pixel to ferromagnetic surfaces.

### 3.2 Charging

A custom charging station (Figure 3a), resembling a large egg tray, charges 25 pixels at once. It is a custom 40 by 31 cm PCB board suspended above a wooden base. Each pixel rests in a 39 mm circular hole which has an exposed conductive rim connected to charging circuitry. When in the hole, the pin on the side of the pixel makes contact with the rim, and the pin at the bottom of the pixel contacts a ground plane. The ground plane is fabricated from copper tape bonded to thin plastic film, which is attached to the charging station wooden base in a way that sections of the strip form very light springs (Figure 3b). Wood spacers suspend the charging board 16mm above the base. The charging circuitry for each hole is a MCP73831T-2ACI Lipo charge controller chip with an LED to monitor charging status.

A pixel begins charging as soon as it is inserted into the circular opening, and it takes approximately 90 minutes to charge. After charging, a pixel will run continuously for 5 to 8 hours, depending primarily on LED illumination time. Note that pixels have no "on and off" switch: as soon as the firmware has power, the radio runs in receive mode and processes data from the base station. This makes the system simple to deploy, but further optimizations like a "sleep mode" could drastically increase stand-by time for charged pixels. We created four of these charging stations, so in practice, all pixels can be left charging until needed.

### 3.3 Base Station

The system uses a single base station (Figure 2a), which is an ArduinoMega microcontroller,bluetooth module HC-05 and a 2.4 GHz NRF24L01+ wireless transceiver board from (https://www.sparkfun.com/products/705) operating as a transmitter with a SMA connector. An external LCW Dipole high-gain antennae is connected to the SMA port. The base station connects to a standard smartphone through the HC-05 Bluetooth radio. The typical workflow in sending data to the pixels involves creating the data packet on the phone app, which is sent to the base station through bluetooth. The Arduino Mega decodes the received packets and sends appropriate signals to all physical pixels using the NRF24L01+ radio. The base station is programmed with calibration routines used for pixel registration. In informal tests, we found the base station could communicate with pixels more than 7 m away without major obstructions like walls.

### 3.4 Firmware and Communication Protocol

During assembly, custom firmware is loaded into each pixel microcontroller using an USB-UART programmer through a 5-pin header on the pixel PCB. The program assigns a unique id, operates the radio in receive mode to continuously listen for data packets from the base station, and when data is received, decodes the data and performs the required action. The firmware also sets the radio power amplifier to LOW level (to reduce power consumption), sets the radio channel, and sets the air data rate to 250 Kbps. For a display to show meaningful content, all the pixels of the display should update synchronously. We did not use a standard mesh network protocol because it introduces a dependency on certain pixels and increases latency as data traverses through the network. Instead, our method is multicast, where all pixels receive the same data packet from the base station at the same time. This means all pixels can be updated simultaneously, which avoids latency during registration or when displaying content across many pixels forming a single display. In addition, there is no dependency on any pixel for communication, so the system is robust if a pixel battery drains before others.

The communication protocol uses a single 26-byte data packet. The first byte encodes one of three commands, and the remaining 25 bytes encodes which pixels must execute the command. The commands are: `activate`, to signal a pixel to illuminate the LED; `register`, to run a pre-determined registration routine of flashing the LEDs at 33Hz for two cycles; and `flash`, to flash the LED in a binary sequence representing the pixel's unique ID. Since the same data packet is broadcast to all pixels, multiple pixels can execute the same command and which pixels should execute is determined by a bit mask. Our current implementation supports 100 pixels, where the position of each bit in a 100-bit binary string indicates whether the corresponding pixel id should execute the command ('1') or not ('0'). For convenience when decoding the packet on the microcontroller, the 100-bit binary string is encoded in a 25 byte hexadecimal string.

Consider a simple example. If a pixel with id '7' receives a packet with first byte indicating `activate` and decodes the remaining bytes to determine the 7th bit is '1', then it will illuminate its LED. Correspondingly, if a pixel with id '11' receives the exact same packet with `activate` in the first byte, but decodes the remaining bytes to find the 11th bit is '0', it will turn off.

Our system can update up to 100 pixels at 20 Hz from the phone, the limiting factor for the refresh rate is Bluetooth communication between the phone and base station. Using our simple broadcast protocol, the base station and pixel hardware is capable of 100Hz updates. For applications that do not require interactive phone input, a one-time configuration packet could be sent to the base station to to control a sequence of pixel updates at 100 Hz.

### 4 SPATIAL LOCATION REGISTRATION

The physical pixels can be arranged in different configurations, like small and large 2D displays, and long 1D displays. Once arranged, the relative location of each pixel and its ID are required to display content appropriately. In a basic form, registration may be accomplished by holding the phone camera to capture all pixels at once, then recording a video while the pixels to execute a flashing registration sequence. A more generalized form of registration, suitable for large, disconnected, or dispersed displays, is to repeat this capture

sequence multiple times by moving the phone camera over the display, pausing when one or more pixels become visible in the frame, and capturing a video of the flashing registration sequence each time. This method reconstructs a single spatial representation of the display in 3D. In essence, the interaction and goal is similar to taking a panorama photo: separate or overlapping portions of the display are captured from different view perspectives, and then "stitched" together to create a single spatial registration for all physical pixels.

In both registration forms, an Android application uses computer vision code to extract the image coordinates and IDs for each pixel, which are combined and stored as a single *spatial frame*. In the generalized form, this is repeated for each capture sequence, with the relative position and orientation of multiple spatial frames determined using the Android ARCore smartphone pose. Combined, a single spatial frame is composed of all the IDs and image coordinates for each physical pixel location captured within the frustum of the smartphone camera along with supplementary pose data obtained from ARCore. For the generalized form of registration, all data is sent to a separate Unity server using GRPC[2] to combine all spatial frames capturing portions of the display into a final spatial registration of the complete display.

## 4.1 Basic registration using single capture pose

This form of registration is sufficient for a display that can fit within the frustum of the smartphone's camera from a medium distance. The `register` and `flash` commands allow for two types of pixel identification techniques: *one-shot* and *sequential*.

### 4.1.1 One-Shot Identification

The approach to extract the image coordinates and IDs for each physical pixel is inspired by Firefly [4]. All visible pixels within the camera frustum are located at once by decoding this pattern, which is encoded as a series of LED flashes.

During video processing, we detect the video frame in which all physical pixels are illuminated as the starting point of the binary sequence. This is converted into a binary image from which brightness and circular contours are used to find a region of interest (ROI) for each physical pixel. A single flash of an LED lasts for exactly 100 ms (approximately 12 video frames). A complete binary sequence can then be reconstructed by analyzing each ROI over time. Finally, the decoded ID and centre of the ROI is saved as a JSON string for further processing and visualization. For a 54 pixel display, the capture time is approximately 2.5 seconds and video processing is less than 1 minute. Capture time remains constant and processing time is linear to the total number of pixels. Our calculations suggest that a 1,000 pixel display would require the same capture time of 2.5 seconds and processing time of about 15 minutes without optimization. Video processing time could be further optimized using GPU acceleration and SIMD instructions on the phone.

### 4.1.2 Sequential Identification

In challenging lighting conditions and display configurations, the one-shot method can fail, so we provide an alternative more robust method with the trade-off of more time to capture pixel flashes. This is a similar approach to what is used in Particle Display System [36]. When the user initiates capture, the base station broadcasts packets to create an initialization time marker sequence, where all pixels are illuminated for 200 ms, then off for another 200 ms. Then, the base station broadcasts packets to request each pixel, one at a time in ascending order, to execute the `register` routine of their assigned ID. The time window for each pixel to flash is 200ms. When the capture is finished, the video is processed similar to above to locate each pixel, with the advantage that the problem is more constrained. Only one pixel will be flashing at a time, and the temporal order provides

---

[2]www.grpc.io/

a degree of error checking. Video processing is similar to *one-shot*, were binary thresholding and frame subtraction is used on the video frames. As an example of performance, it takes approximately 38 seconds to process the recorded video to obtain all image coordinates and IDs for a 54 pixel display. To get the best performance, the phone should be held stationary.

## 4.2 Generalized registration with multiple capture poses

The single pose registration is sufficient when the physical pixels can be framed within a single camera view that is held roughly parallel to the plane spanning them. However, if they are distributed over a wider area or have non-planar shapes, multiple captures need to be utilized to reconstruct their spatial relationships. The sequence of captures, along with pose information from ARCore, allow us to build a 3D representation of the physical pixels using an incremental optimization technique over the parameter space of each pose and image coordinate for each captured frame. The time to register a large display is proportional to the number of single frame captures needed for reconstruction, where each capture uses the method described in Section 4.1.

As a starting point, we frame this as a variation on a structure from motion (SfM) [30] problem found in large scale computer vision tasks, with two added assumptions: 1) the correspondences between physical pixels are known and 2) the poses for each capture is approximately known with noise. This brings the total parameter space for each capture with $N$ detected physical pixels to be equal to $\Phi = 6 + N + 2N$ parameters: 6 parameters representing pose, $N$ parameters representing the distance $z$ for where each physical pixel lies on the ray projected from the camera, and $2N$ parameters that represent the image coordinates for each captured physical pixel. To limit the scope of the parameters space into manageable chunks, we formulated an incremental optimization routine that iterates over each pair of captured frames in three separate phases, where each phases is responsible for updating only a subset of the parameters.

Phase 1: Image coordinate projection. This initial phases utilizes assumption 1 and 2, that we know correspondences between physical pixels and we know the approximate pose for the captured frame. For each pair of captured frames, we project the recorded image coordinates out into a shared world coordinate space to find the optimal value $z$ for each physical pixel that minimizes the distance between the projected 3D world points from one frame to the other. This results in an initial guess on where each physical pixel is located in the world.

Phase 2: Pose adjustments. Based on the results of *Phase 1*, we have an initial guess of were the physical pixels are located in a world coordinate frame of reference. In *Phase 2*, we refine the 6 parameters that represent the pose of each captured frame. For each pair of captured frames, we find the optimal pose that minimizes the distances between each pair of corresponding 3D points representing the physical pixel. This results in further refinement of the captured frames' poses and 3D points of the physical pixels.

Phase 3: Image coordinate refinement. In the final phase, we again project the captured image coordinates into a world frame of reference utilizing the recovered $z$ values from *Phase 1*. We then iterate over each pair of captured frames and directly refine the image coordinates by minimizing the distances between each corresponding 3D point.

All three phases are applied to each pair of captured frames, where $K$ captured frames give $K^2/2$ iterations. Each phase uses a non-linear least square Levenberg-Marquardt [22, 27] algorithm to minimize their cost function. Further refinement can be accomplished through multiple iterative applications of our optimization routine that utilize parameter regularization to further constrain the dimensionality of the overall parameter space. The final result gives the 3D points for each physical pixel, with accurate sense of scale

and space. We use these final 3D points to create an accurate 2D image of the physical pixels by projecting them back into image space using an orthographic projection that encapsulates the entire physical display.

## 5 MAPPING AND RENDERING IMAGERY

Mapping images onto an ad hoc reconfigurable display is not always straightforward since the pixels can be arranged in an arbitrary fashion. In this section, we describe different methods to display content on the ad hoc display which includes directly controlling the physical pixels from the phone to create animations and mapping existing binary images to it. We also describe our interactive layout assistant that guides the user to optimally place the pixels when content is known beforehand.

*Interactive Display and Animation* — This mode allows the user to directly control the display pixels using the phone. A phone app shows the spatial map of the display on which the user can directly draw and create animations. As the user draws an image, the input events can be saved and played back as animations. The user can also create individual frames that can be saved and played back at a specified frame rate. The pixels update in real-time to reflect the drawings and animation frames.

*Image Rendering* — Rendering a given bitmap image onto registered positions of physical pixels, uses a simple proximity mapping. First, the image is downsampled and binarized. Then, given an image position and scale in the physical display, the closest physical pixel to each image pixel is determined. For manual control, the phone app enables the user to position and scale each image in a set with a live preview on the display. These adjustments are saved, and used to display each of images at the configured locations and scales to create the dynamic display.

*Optimal Image Mapping* — Rather than manual positioning, an optimal location can be found. Given an image, a stochastic algorithm iteratively places the image at different positions within the display until it finds an optimal position. Optimal corresponds to minimal error computed as the sum of Euclidean distances between the image pixels and its corresponding nearest display pixel.

### 5.1 Interactive Layout Assistant

We also created an interactive layout assistant guides a user in placing the pixels at optimal locations, when the set of images to be shown on the display is known beforehand. The pixels are initially spread out in some arbitrary configuration and a quick pixel registration is performed to obtain the initial layout of the pixels. The user chooses a set of images from an image gallery and sent to the python server, which generates an layout for the display. A python server binarizes all the received images, and a resultant binary image is obtained by performing a boolean OR with all the binary images. The resultant high resolution binary image is down-sampled using k-means clustering, by setting the cluster size equal to the number of physical pixels present in the initial layout. The output cluster centres gives the optimal location to place pixels.The suggested pixel location values are scaled to the camera preview frame size and sent to the phone.

The pixel locations obtained from the server are overlaid on the camera preview as circular guides to assist the user in arranging the pixels according to the suggested display layout. The layout assistant automatically turns on four pixels acting as anchor pixels, which should be placed within the red circular guide overlaid (Figure 4 b). Once the anchor pixels have been placed in the appropriate locations, the remaining pixels are placed within the white circular guides (Figure 4 c). The white circular guides are always displayed with respect to the anchor pixels, such that even if the phone moves around, they are re-positioned with respect to the anchor pixels. This is achieved by continuously tracking the anchors pixels in

real-time and computing the homography with respect to its initial position. The new position of the white circular guides are obtained by warping it with the computed homography. The user can add or reduce the number of pixels to the display, and the layout assistant generates a new layout for it. Once the pixel locations are fine-tuned by placing them within the white circular guides (Figure 4 c), a quick pixel registration is performed to obtain a spatial map. Now, the display can cycle through the dictionary of images (Figure 4 d,e).

## 6 APPLICATIONS USING DIFFERENT CONFIGURATIONS

This section describes a broad collection of possible real world applications for the Scatterpixels system using different display geometries and environment locations. The emphasis is on non-permanent temporary installations that might be typically created on a smaller scale by non-professionals. The display applications are typically designed to exist for a short time withing the 5 to 8 hour battery life of the pixels, such as an hour up to an afternoon or evening.

Our goal in presenting many different applications is to demonstrate the versatility of the system, and show it can easily be reconfigured into different display configurations with minimal setup time. Note the same set of pixels are reconfigured to form all applications, so these demonstrations serve as a simple validation of system reconfigurability in terms of pixel density, shapes, sizes, and locations. We note that the Scatterpixels system was designed with a vision of enabling display applications using hundreds of pixels. As a research proof-of-concept system, we have a limited number of physical pixels to implement and illustrate the examples below. However, the system and application concept can be expanded to much larger scales. An accompanying video shows a large subset of these demonstrations, including how the implemented display changes over time or animates.

### 6.1 Individual Pixel Displays

The simplest configuration is using each pixel as single bit visual indicator to convey a state or status which changes over time. Each pixel can be individually controlled by ID, and could optionally be associated with a specific person or object.

*Facilitating Games with Large Groups* — Pixels could be distributed to people attending a party, banquet, film showing, or other event. Without knowing which pixel ID is held by each person, the activation status of a pixel could still be used to form random teams to compete in a game, or enable ice breaker activities, like random groups of an audience singing parts of a song. This can be achieved by selecting a random subset among all pixels IDs that were distributed, and flashing those pixels together with instructions by the organizer to form a team, or as a cue to sing during the performance. The pattern of flashes could even be used to indicate a team leader among the subset, or divide the subset into different musical parts.

*Personal Notification* — Pixels can be used to generate visual notifications to track the progress of an activity or convey basic information like navigation instructions while walking. The flashing frequency of the pixel can indicate the progress of a microwave timer or a reminder for an upcoming appointment. Custom flashing patterns can be used to provide simple navigation instructions, like slow flashing for a right turn, fast flashing for a left turn, and solid red to indicate that the destination is reached.

### 6.2 One-Dimensional Displays

Pixels can be arranged along a path or in lines to form one-dimensional displays. Once registered, animated patterns can convey information like direction, activity, and time along the display.

*Meeting Location Display* — Pixels can be arranged in line along floors through corridors, and show animations to help attendees locate a meeting room and remind them about the time remaining

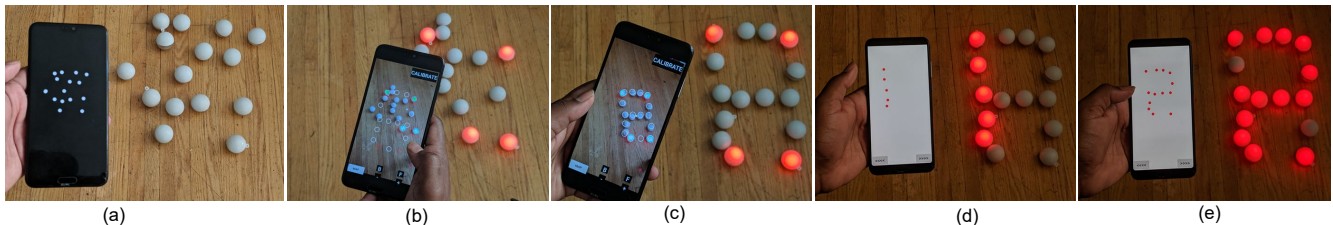

Figure 4: Interactive layout assistant: (a) initial layout with pixel arranged in random configuration; (b) guide overlay with red circles showing where to place illuminated anchor pixels; (c) all the pixels are arranged within the white guidelines; (d,e) examples of imagery;

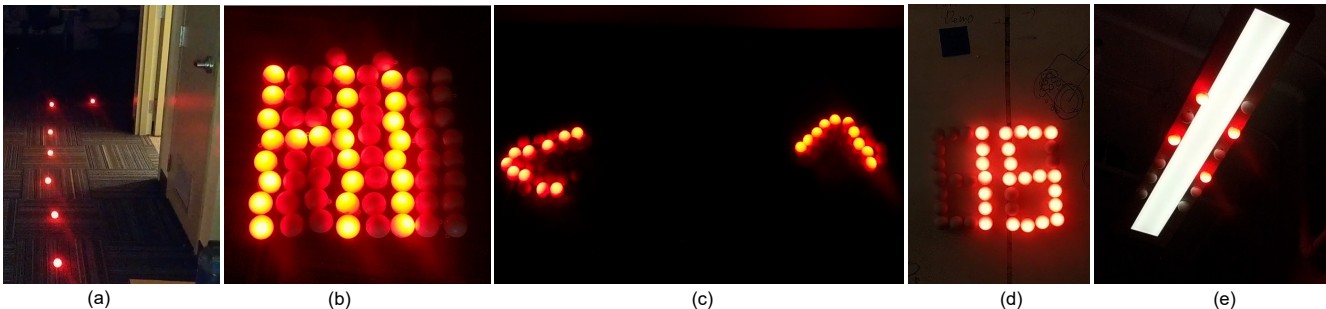

Figure 5: Applications: (a) 1D guidance display (b) 2D floor display showing 'HI' (c) 2D floor queue displays ; (d) 2D White board timer display; and (e) 2D Ceiling displays

before the meeting starts. For example, each pixel can light up at time along the path to indicate the path direction(Figure5a). This animation can speed up to indicate the meeting is starting soon, all illuminate when the meeting has started, and then flash together for a few minutes to hurry people to the meeting. Once its too late to join the meeting, then can all turn off.

*Wait Queue Indicator* — At an event, the pixels can be arranged in multiple lines beside different queues to indicate the waiting time or type of queue using different patterns and animations. This would allow people to choose the correct or optimal queue.

### 6.3 Two-Dimensional Sparse Displays

Multiple pixels can be distributed throughout a space, and then spatial registration methods can enable individual pixels to communicate information about their location.

*Targeted Class Participation* — At the start of a large lecture, each student can be provided with a pixel with a known ID. This could allow the teacher to call on specific students based on records of past participation, to award prizes, or to split students strategically who are sitting near each other for a classroom activity. The association between pixel ID and student could be done using a variation of our registration method using a very high resolution camera in the lecture hall to capture the pixel ID flashes and an enhanced algorithm to associate the pixel location with the recognized face of the student.

*Location Indicators* — Pixels could be placed throughout a temporary setting like a banquet hall, weekend craft workshop, or farmer's market, and the illuminated pixels used to indicate an area or item. For example, to guide a guest to an open table, assist a workshop participant in locating a specific material, or highlighting sales items to a shopper. More than one pixel could be used at each location to convey more information based on illumination pattern, such as stock level or urgency.

### 6.4 Two-Dimensional Dense Displays

Multiple pixels can be placed together on the floor, or attached to a metal wall structure like a whiteboard or architectural panels, or

even to metal fixtures and elements in a ceiling. The pixels can be placed randomly or formed into specific shapes, and they can be separated into different clusters.

*Floor Sign for an Event* — The pixels can be arranged on the floor or ground to create an ad hoc display. Once calibrated, the display could show a welcome note during a party as an example (Figure5b). The people in the party could also use their phones to create drawings, which could provide additional amusement for the people.

*Exam Timer or Sport Scoreboard* — During an exam in a large lecture hall, pixels can be arranged on a whiteboard to indicate the time remaining(Figure5d). This could be conveyed using numerals, shown at optimized positions on a randomly assembled cluster of pixels, or on a more intentionally laid out display using the layout assistant. Alternatively, the time left could be communicated through more abstract patterns. A similar, but equally compelling application is creating a scoreboard at an ad hoc location for an amateur or informal sports event.

*Queuing Signs* — As an extension of the event queue example above, multiple clusters of 2D pixel displays could be placed at the beginning of different queues. Each cluster could show symbols or patterns to indicate which queue is open or slow, and suggest alternate queues. For example, displaying shapes like an up arrow for open, flashing for slow, or left and right arrows to suggest the direction of other queues(Figure5c). This could be useful to direct crowds at large festivals, open markets, or even in emergency response situations.

*Hanging Displays* — Some pixels have a tab with a hole extending from the case, which can be attached to strings or hooks to create different types of hanging displays. With many pixels, this could be an alternate form of vertical 2D display, like a sign. Or it could be used for decorative, ambient effects, such as outside in a garden like the NetworkedPixels [6] project.

*Ambient Effects* — Pixels arranged on the ceiling or walls(Figure5e) can be used for ambient effects, like calming pat-

terns at yoga retreat, or accentuating dance music at a festival. They could be attached arbitrarily to any available ferromagnetic elements, like light fixtures or steel structural beams.

## 7 DISCUSSION

We discuss current limitations in our system and future enhancements to further expand the capabilities and possibilities for this type of ad hoc display.

*Pixel Size and Display Resolution* — The 40mm physical size of each pixel means that the effective display resolution is limited unless viewed from a great distance. Moving to a higher frequency communication would reduce the antenna size and effectively reduce the pixel size.

*Display Colour* — For simplicity, we use a red LED to create monochrome displays. Each pixel can be easily extended to support a RGB LED to create a colour display, in fact our LED daughter board is designed to support the pins for an RGB LED. The challenge is the impact on power requirements to drive an RGB LED, and a significantly expanded communication protocol that increases from 1-bit to activate an LED to many bits per pixel to specify the colour as well. Another related way to expand the fidelity of the display is to add more bits to select an intensity level of a single colour LED though PWM. A useful range may be as little as 2 or 3 bits for 3 to 7 intensity levels. Enabling reduced intensities would also reduce amortized LED power consumption.

*Run Time* — The run time of a pixel could be extended with more efficient hardware and software. For example, adding a "sleep mode" that only checks for base station packets every minute before waking up could drastically increase stand-by time for charged pixels. However, the battery-to-weight ratio will always impose some limits on maximum run time. Other approaches like battery-free pixels that harvest power from ambient energy sources [9] could eliminate this run time ceiling. We initially experimented with an RFID-based approach using a "Rocky 100" [14] chip to control an LED and harvest power from RF signals sent from the RFID reader antenna. However, the power harvesting capabilities and communication link to be very unreliable, and the large antenna limited how closely pixels could be packed.

*Scalability in Terms of Number of Pixels* — Using the current base station setup and communication protocol, we can control up to 124 pixels. This number can be increased using the same radio by partitioning hundreds of pixels into sets of 124, sending commands to each set of pixels in batches, and using an offset update signal sent synchronously to all the pixels to make it appear like all pixels update at the same time. This method will reduce the frame rate of the display. Another approach is to modify the base station to support more transmitters. All transmitters can work in parallel, controlling all sets of 124 pixels simultaneously, without compromising the frame rate. However, this requires a fixed association between pixel and transmitter which increases complexity and cost.

*Pixel Tracking for Registration* — The optical tracking of the pixels using a RGB camera does not work reliably in all lighting conditions. Using non-visible light methods for registration, such as the phone NFC reader to scan NFC tags attached to each pixels, or the phone IR receiver to decode a pixel ID flashing sequence transmitted from an IR led in the pixel.

*Interactive Pixels* — Our system works as an output device by controlling a LED based on the signal received from the base station. But,the pixels can be instrumented with additional sensors to sense touch, motion,light, sound and other sensors. The sensed information can be used to directly manipulate the pixels state or the information shown by the display. Instead of instrumenting all the pixels with sensors, we could design specialized "super" pixels, which are instrumented with sensors. For example, a microphone embedded into the pixel can listen to the user's question and show

an appropriate response on the display, or like Particle Display [36] pixels instrumented with an accelerometer allows to interact directly with it through motion.

*3D Displays* — In principle, these pixels could also be used to create 3D displays. For example, pixels wrapped around a cylinder, or even pixels hanging in a cluster. Our registration method supports the basics of finding pixel locations in 3D, but we would need to relax and refine optimization assumptions, and possibly require more images and more guidance to perform a full 3D registration. Displaying images on a clustered 3D configuration would will require a significant extension to our image fitting algorithms.

*Automatic Layout* — The process of physically laying out pixels to create a display can be made more convenient by using some time of "paint roller" loaded with pixels. Pixels can also be loaded onto a cylindrical metal stack which can spit out the pixels, when a button is triggered. This process can be completely automated by using a robot programmed with the layout configuration to place the pixels at appropriate locations.

## 8 CONCLUSION

We presented Scatterpixels, a system using custom-built wireless LED pixels that can be arranged in multiple ways to form different kinds of ad hoc displays. Unlike previous work, we describe a full end-to-end solution including hardware, software, and user interfaces for setup. Our individual pixels are simple to set up in many different layouts, and can be conveniently controlled from a smartphone. We developed a comprehensive set of spatial registration methods to accommodate different display configurations, and we provide methods to map content to the displays, including an interactive layout assistant to guide the optimal placement of pixels when expected display content is known. We show how these pixels enable flexible display configurations ranging from one-bit indicators, to 1D lines, to different 2D shapes, clusters, and surface orientations. Our work is a step towards a grander vision, in which individual display pixels are even smaller, powered wirelessly, and inexpensive enough to be "painted" on surfaces, scattered across floors, and embedded in building materials — creating a future where pixels, and displays of all shapes and sizes, can literally be everywhere.

### ACKNOWLEDGMENTS

This work made possible by NSERC Discovery Grant 2018-05187, Canada Foundation for Innovation Infrastructure Fund 33151 "Facility for Fully Interactive Physio-digital Spaces," and Ontario Early Researcher Award ER16-12-184.

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
