# OpenReview forum: "Scatterpixels: Ad Hoc Reconfigurable Physical Pixel Displays"
_graphicsinterface.org/Graphics_Interface/2023/Conference — GI 2023_

### Official Review · Reviewer_vjjD · 2023-01-06
**Well written paper, clear and potentially impactful contribution even if narrow.**

**Rating:** 8
**Confidence:** 4

**Review:**

This paper presents Scatterpixels, an end-to-end system that allows users to quickly deploy and calibrate a set of wireless LEDs that can form monochrome images and animations. The system allows users to deploy the LEDs in both small and large arrays, and also provides optimizations to help users guide the physical placement of LEDs. The interface is simple, but allows users to directly manipulate the state of the light, or upload images or a set of images to control the display.

Overall, I found this paper well written and easy to read. While there has been quite a bit of work on distributed displays, I appreciated that the paper was quite clear that their contribution was the consideration of the end-to-end technical aspects in putting such a system together. I appreciated this detail, as it makes it very clear what the aim of the paper is, and while I think the contribution is narrow, it is still a strong contribution and I would argue for publishing it at GI. The potential applications in section 6 also highlight why this type of approach and contribution might be quite impactful and find utility across a number of different use-cases and user groups.

I appreciate the detail that went into section 3. Normally, implementation sections of novel devices or approaches are overly brief, causing researchers to struggle to replicate such systems. In contrast, the level of detail here (including specific model numbers, as well as detailed processes) support the contribution of the end-to-end technical discussion. I also appreciate the inclusion of source code, design files and schematics on publication.

The video, along with section 4 highlight the user process for setting up the system, and it does seem like it would be quite easy. I think a larger mobile device might give a bit more control, but that's a simple fix. It's arguably beyond the scope of this submission, but I would be interested in seeing how novices would use this system. In some of the use cases (e.g., a scoreboard for a game), the end users might not have any technical background - what would that workflow look like, and does there need to be a set of presets for common use cases like that? One other minor criticism around the user interaction is the lack of support for video. It looks like the system allows for uploading sequential frames, but could it support uploading a video directly? It doesn't seem like a very technical challenge, but it is one that I think would help with the end-user experience/functionality.

Minor: I think there's a typo, 3.1, first para: 'and _60 angle of half intensity'.

Overall, I think this paper would make a great addition to GI 2023.

---

### Official Review · Reviewer_jP28 · 2023-01-12
**An impressive engineering contribution**

**Rating:** 9
**Confidence:** 4

**Review:**

This paper presents scatterpixels: individually addressable and reconfigurable LED displays.

Overall, the paper does have some comparable technologies with similar types of applications, however, the authors acknowledge this and cite this work. I trust the notion that no other work has gone to the extent of providing a well documented and clear engineering contribution. I believe the there is sufficient detail in this work for people to build their own scatterpixels.

The authors also provide thoughtful analysis of energy consumption and battery life, scalability, and natural extensions (e.g., multi-color lights).

I have little to critique in this paper, and so I would very much like to see it at GI.

There are a couple of sentences that need to be revised.

1) in the abstract the sentence “The fabrication and hardware… and more than 70 assembled.” Is not a complete sentence and needs to be rewritten.

2) the last sentence in the Interactive Pixels section of the Discussion needs to be rewritten.

---

### Official Review · Reviewer_xKXf · 2023-01-17
**Compelling system with an unclear contribution**

**Rating:** 5
**Confidence:** 4

**Review:**

This paper presents a system for calibrating and controlling ad hoc displays configured from voxels. An AR-based app allows a dispersed set of voxels to be spatially registered. The work also demonstrates a number of various ways of interacting with the display dynamically.

I quite enjoyed reading this work. The system demonstrated is interesting an clearly the result of a considerable effort. However, the contribution is a little bit difficult to pin down. The paper does a very thorough job of covering a diverse range of related systems, but struggles to clearly differentiate the research contributed here. The summary sentence of the last paragraph of the introduction is rather vague and sounds much like it is saying 'there is nothing really novel here but we've done everything better." I would encourage the authors to improve this statement to be a more clear about what they are offering. Surely this system has some nuggets of original work that can be claimed!

After reading further on, it seems the main contribution is in the registration method, which allows calibration of a large number of widely dispersed voxels. This could be made more clear in the intro as well as the description of the method. The paper also demonstrates a wide variety of use cases that the app enables. These come across nicely in the video. There seem to be some new interactions explored here - again it would be great if this can be clarified by the authors. It would improve the contribution if these interactions could be organised into a more concrete design space. A framework that describes the different types of interactions possible, along with the requirements for each, would be useful in itself. I am also surprised that all of the examples seemed to involve planar configurations of the voxels. Having more 3D configurations, for instance voxels placed at different heights in a room, would help demonstrate the utility of the registration method.

Overall I'm a bit on the fence with this work, since it seems like the contribution is not yet well defined, and there is more yet to be done (e.g. a well organised taxonomy or design space of the interaction possibilities).

---

### Meta-Review · Area_Chair_RPeu · 2023-01-17

**Recommendation:** 7
**Confidence:** 4

**Metareview:**

I was somewhat cautious in my independent review of this work. However, the other reviewers are more convinced that the work presents a substantial engineering contribution that is ready for publication. They mention some strengths of the work including a complete and refined implementation, and a thorough description of the technical details, which will help allow future researcher to create similar implementations.

Given the reviewers' enthusiasm for this work I recommend it be accepted and shared at GI. I suggest the authors consider the suggestions for improvement in drafting their final version. I also encourage a thorough proofreading of the paper to address several grammatical issues including and beyond those notes in the reviews.